# Abscisic Acid Inhibits Asymbiotic Germination of Immature Seeds of *Paphiopedilum armeniacum*

**DOI:** 10.3390/ijms21249561

**Published:** 2020-12-15

**Authors:** Xin Xu, Lin Fang, Lin Li, Guohua Ma, Kunlin Wu, Songjun Zeng

**Affiliations:** 1Guangdong Provincial Key Laboratory of Applied Botany, South China Botanical Garden, Chinese Academy of Sciences, Guangzhou 510650, China; xuxin17@scbg.ac.cn (X.X.); linfang@scbg.ac.cn (L.F.); lilin@scib.ac.cn (L.L.); magh@scib.ac.cn (G.M.); 2University of Chinese Academy of Sciences, Beijing 100049, China; 3Key Laboratory of South China Agricultural Plant Molecular Analysis and Gene Improvement, South China Botanical Garden, Chinese Academy of Sciences, Guangzhou 510650, China

**Keywords:** *Paphiopedilum armeniacum*, ABA, germination

## Abstract

*Paphiopedilum armeniacum* is a rare orchid native to China with high ornamental value. The germination of *P. armeniacum* seeds is difficult, especially for the mature seeds, which is the major limitation for their large-scale reproduction. This study explored the reasons for seed germination inhibition from the aspects of the important plant endogenous hormone—abscisic acid (ABA). The major endogenous hormone contents of seeds were determined at different developmental stages. The ABA content was 5.8 ng/g in 73 days after pollination (DAP) for the immature seeds, peaked at 14.6 ng/g in 129 DAP seeds, and dropped to 2.6 ng/g in the late mature stage of the 150 DAP seeds. The reduction of ABA content in the mature seed suggests a possible contribution to the increased expression of *CYP707A*, an ABA catabolism gene. The germination rate of the immature seeds was reduced to 9% from 69% when 5 μg/mL ABA was added to the Hyponex N026 germination medium. The result showed that ABA can inhibit the germination of *P. armeniacum* immature seeds. However, for the heavily lignified mature seeds, reduction in endogenous ABA level does not result in an increase in the germination rate. Lignin accumulation in the seed coat imposes the physical dormancy for *P. armeniacum*. In summary, the germination of *P. armeniacum* is regulated by both ABA and lignin accumulation.

## 1. Introduction

The *Paphiopedilum* genus belongs to the family of Orchidaceae, containing 107 species throughout the world, and are distributed across southern India, Nepal, Bhutan, north-eastern India, Burma, southern China, Hong Kong, South-east Asia, the Malay Archipelago, the Philippines, New Guinea, and the Solomon Islands [1,2,3]. The *Paphiopedilums* are attractive and popular in the horticulture market due to the unique pouch-shaped lip of the flower and their abundant colors. Unfortunately, the *Paphiopedilums* are left in a dangerous situation with their supply decreasing sharply because of habitat destruction and excessive collection. In an effort to conserve this endangered species, the Convention on International Trade in Endangered Species of Wild Fauna and Flora placed the entire wild *Paphiopedilum* species on the list to forbid international transactions [4].

The conventional propagation of *Paphiopedilum* orchids using axillary buds division from the mother plant is unproductive and time-consuming [5]. Asymbiotic germination is the most common and efficient approach for the large-scale propagation of *Paphiopedilum* [1]. However, the germination of fully mature *Paphiopedilum* seeds is often difficult. Studies on the embryo development and germination of *Paphiopedilum* showed that *Paphiopedilum* seeds have an optimal germination stage before reaching maturity: The germination of *P. armeniacum* was about 65% in 120~130 days after pollination (DAP) seeds, while the 180 DAP seeds failed to germinate [6]. In 180 DAP seeds, *P. wardii* and *P. hangianum* had the highest germination (65% and 73%, respectively), while the germination was reduced to 20% in 300 DAP seeds [7,8]. The germination of 270 DAP seeds of *P. delenatii* and *P. callosum* reached above 90%, while The germination of 300 DAP seeds of *P. callosum* was only 20% [9,10].

Orchid seeds have a unique characteristic: a capsule containing thousands of tiny seeds of about 0.05–6.00 mm with various shapes [11]. The mature orchid seeds are composed of undifferentiated embryo covered with testa and no morphologically differentiated endosperm and cotyledons, which is different from most flowering plants [12]. The research about the failure of orchid germination focused on some hypotheses: (1) morphological dormancy caused by underdeveloped embryos; (2) physiological dormancy caused by inhibitors accumulated in mature seeds; and (3) physical dormancy resulting from thick impermeable testa [13]. Many studies have reported that abscisic acid (ABA) is an inhibitor of seed germination and plays an important role in inducing seed dormancy [14,15,16]. Early research found that mature terrestrial seeds of *Epipactis helleborine*, an orchid species with poor germination, contain five times more free ABA than *Dactylorhiza maculate*, an orchid species with higher germination [17]. The orchid species *Calanthe tricarinata*, *Cypripedium japonicum,* and *Cypripedium formosanum* have relatively high ABA in their seeds, and some of the treatments to reduce the ABA content in *Calanthe tricarinata* and *Cypripedium japonicum* mature seeds were shown to improve seed germination [18,19,20].

At present, the research of *Paphiopedilum* seed germination is still mainly based on the observation of seed morphology. *P. armeniacum*, a native orchid species to the Yunnan province of China, has been praised for its high ornamental value. In a previous study, key anatomical features in embryo development were shown to be associated with the ability of embryos to germinate in vitro [21]. The fertilized egg is formed around 45 DAP. At 87 DAP, it undergoes longitudinal and lateral division of the cell to form a preliminary globular embryo with the suspensor also beginning to degenerate. After further development around 129 DAP, the final mature embryo is still ellipsoidal without morphological differentiation. In general, the highest germination rate occurs when the globular embryo is formed and the suspensor has not fully degenerated. Similar embryogenesis and development have also been explored in *P. barbigerum*, *P. appletonicnum*, *P. concolor*, *P. hirsutissimum*, *P. delenatii*, and *P. helenae*, although different *Paphiopedilum* embryos have different developmental and maturity cycles [22,23,24]. Transcriptome and lignin analysis studies further revealed the large amounts of non-methylated lignin that accumulated during the seed maturation of *P. armeniacum*, which negatively correlates with the germination performance [25]. However, the role of ABA and other important plant hormones are still uncertain. A better understanding of seed physiology will provide valuable information concerning the propagation and help with the conservation of those hard-to-germinate species. The goal of this study is to investigate the changes in endogenous hormones during seed development and the effect of ABA on the asymbiotic germination of *P. armeniacum*.

## 2. Results

### 2.1. P. armeniacum Seed Is Composed of Only an Undifferentiated Embryo and One Layer of Heavily Lignified Seed Coat

Orchids typically produced capsules with dust-like seeds inside (Figure 1(A1–A3)). As *P. armeniacum* seeds developed, they gradually changed from a whitish color to brown and black (Figure 1(A1–A3)). Before 94 DAP, the color of the seeds in the capsules was white, and after 94 DAP, the color began to turn brown. At 150 DAP, the seeds became dark brown and black.

Scanning electron microscopy was used to collect information on seed size and seed coat structure. The individual seeds were spindle-shaped, and the surface of the seed coat was composed of numerous testa cells (Figure 1B). The testa cells were large and rectangular in the middle, and small and round at the ends. This pattern probably resulted from a slow cell division in the integuments during embryo development [26]. During seed development, the seed size increased before maturity at 115 DAP because of cell division and elongation. After maturation, the size began to decrease significantly due to the dehydration-induced shrinkage.

To get an overview of the embryo structure inside the seed coats, the seeds were pretreated with sodium hypochlorite to loosen the rigid seed coat structure [27]. The ellipsoid seed embryo was located in the middle of the spindle-shaped seed, and the seed coat was relatively thin in the middle of the seed compared to both ends (Figure 1C). Unlike the model plants such as *Arabidopsis*, these embryos consisted of cells with similar size and no structural polarization. The size of the ellipsoidal embryos increased from 73 to 115 DAP. From 115 to 150 DAP, the embryo size remained relatively constant (Table 1). The embryos occupied a small proportion of the space inside the testa (20 to 30%), leaving a large amount of air space. Previous studies on histochemical analysis showed that the seed coat is a one-layer structure and consisted of the hydrophobic lignin, serving to protect the embryo [21]. The mature seed coat cells are all vacuolated. At maturity, the cells become dehydrated and collapse into a thin layer.

### 2.2. P. armeniacum Seed Germination Rate Increased During the Early Stages of the Seed Development and Decrease Dramatically after 94 DAP

The asymbiotic seed germination for different stages of seed development is shown in Figure 2. The seeds collected at 73 DAP showed a germination rate of 49%. As the seeds developed, the germination rate increased and reached a maximum of 69% at 94 DAP. As the seeds continued to develop, the germination rate decreased sharply and was reduced to 15% in 115 DAP seeds. After 136 DAP, the seed germination rate was as low as 10%, while in 157 DAP seeds, it dropped to a low of 2%. The germination characteristics of the seeds also appeared to be correlated with the morphological observation of seed color. The period of high seed germination was when the color of the seed was still whitish, while poorer germination appeared to be associated with darker seeds. This observation provides a certain convenience for asymbiotic germination and sowing. The physical appearance of the seeds can be used to judge their maturity.

### 2.3. Determination of Major Endogenous Hormones Level in Developing Seeds

The coordinated interaction of endogenous hormones such as indole-3-acetic acid (IAA), ABA, and gibberellins (GAs) is known to play certain roles for seed germination. The content of four major endogenous hormones during seed development was measured by high-performance liquid chromatography tandem mass spectrometry (HPLC-MS/MS; Figure 3 and Appendix A). The ABA content was maintained at a relatively low level of around 5.8 ng/g at the early stage of seed development and peaked at 14.6 ng/g in 129 DAP seeds (Figure 3A). A similar trend was found for *Cypripedium formosanum* and other orchid species [19,20], which indicated that ABA level remained low during early seed development and rose rapidly mid-way during seed maturation. The increased amount of endogenous ABA may contribute to the germination inhibition before the embryo’s maturation. In addition, as an important abiotic stress hormone, ABA may play an important role in the dehydration tolerance of seeds. However, as the seeds matured further, the ABA level dropped to 2.4 ng/g in 150 DAP seeds. At 150 DAP, the role of germination inhibition is possibly served by the accumulation of lignin instead of ABA [25]. ABA level peaked at 129 DAP and decreased after seed coat lignification was accomplished.

IAA content increased progressively in the early stages of seed development (Figure 3A). The IAA level was 17.5 ng/g in 73 DAP seeds and peaked at 19.9 ng/g in 94 DAP seeds. The level was reduced to 17.2 ng/g in 115 DAP seeds and 10.9 ng/g in 129 DAP seeds. In 150 DAP seeds, it reduced to the lowest level at 4.3 ng/g. This result showed that IAA content was higher in the early stages of the seed development compared to mature seeds. Previous research revealed that the globular embryo of *P. armeniacum* was formed by 94 DAP. The high content of IAA may facilitate the formation of zygotic embryos to globular embryos.

Gibberellic acid (GA_3_) level remained relatively low throughout the whole development of the seeds (Figure 3B). There are other bioactive gibberellins such as GA_1_ or GA_4_ besides GA_3_, so the other types of GAs in *P. armeniacum* may play an important role in seed development. The content of trans-zeatin riboside (TZR) is high in the early stage of seed development and is later decreased and maintained at a low level during the later stages (Figure 3B). As a cytokinin, TZR plays an important role during the cell division of zygote embryo formation to spherical embryo formation [21,28].

### 2.4. Gene Expression Related to Hormone Biosynthesis

The hormonal action of ABA is precisely controlled by the balance between its biosynthesis and catabolism. One differentially expressed gene (DEG) related to ABA biosynthetic pathway and two DEGs in the ABA catabolism pathway were discovered.

In the metabolic pathway of ABA (Figure 4), the expression of *NCED*, encoding an important rate-limiting enzyme for ABA synthesis, increased as the seeds matured and plateaued at 108 DAP. The expression level remained high at the later stages of seed development from 108 to 150 DAP. The expression of the ABA catabolism enzyme ABA 8’-hydroxylase (CYP707A) increased significantly with the maturity of the seeds. Combined with the endogenous ABA level change in the developing seed, there is a possibility that the increased expression of *PaNCED* contributes to the ABA accumulation in the early stage, while increased expression of the ABA catabolic gene *PaCYP707A* is responsible for the ABA level reduction in mature seeds of *P. armeniacum*.

In the metabolic pathways of GA, the gene expression levels of *GA20OX* and *GA3OX*, the important enzymes for GA synthesis, increased with the development of seeds. However, the gene expression level of catabolism-related gene *GA2OX* was higher at 66 DAP than later periods, which remained low with little fluctuations.

### 2.5. Exogenous ABA Treatment Inhibit Immature Seed Germination

To investigate the role of exogenous ABA on seed germination, 5 μg/mL ABA was applied to the asymbiotic germination medium, and the germination performance was recorded. The results showed that ABA inhibited the germination performance by reducing embryo expansion and testa rupture of 94 DAP immature seeds. Although the attached seed coat in the control group (Figure 5A) did not completely fall off, most of the seed embryos swelled and broke through the seed coat, forming an obvious white protocorm. In the treatment group with ABA added (Figure 5B), most of the embryos also swelled but did not break through the seed coat. The seed germination of the ABA-added group was significantly reduced to only 9% (Figure 5C) compared to the 94 DAP seed germination of 69%. This result showed that exogenous ABA did in fact lead to the inhibition of immature seed germination.

### 2.6. Reduction in Endogenous ABA Has no Significant Effect on Mature Seed Gemination

To investigate the role of endogenous ABA on mature seed germination, 94 DAP immature seed capsules were treated with fluridone, an inhibitor of ABA biosynthesis, once a week for a month. At 124 DAP, the ABA content was measured, and the results showed that fluridone treatment reduced the ABA level to 2.8 ng/g compared to 9.4 ng/g in the control (Figure 6). However, there was no significant difference in the germination rate between the seeds that contained low ABA level (2.8 ng/g) and those that contained high ABA level (9.4 ng/g), as shown in Figure 6. This suggests that the asymbiotic seed germination was not improved when the ABA content was reduced in mature seeds.

## 3. Discussion

*P. armeniacum* is a rare orchid native to China with high ornamental value. To help improve the conservation efforts and commercial production of this endangered species, a fundamental understanding of the reproductive biology of *Paphiopedilum* is very important. Basic knowledge of embryo and seed development can aid in the design of experiments that can help explain how to improve asymbiotic seed germination in large scale production. The embryo development and germination pattern of *P. armeniacum* are unique compared to that of other flowering plants (Figure 7). The embryos are poorly developed and consist of cells of similar size and no obvious structural histodifferentiation [29]. Embryo germination gives rise to a tubercle structure called a protocorm, from which the shoot apical meristem subsequently differentiates [30]. There are no necessary signals and nutrients to support histodifferentation before germination occurs because of the lack of an endosperm. This delayed histodifferentiation ensures that only the seeds that arrived at the proper conditions can start the vegetative establishment.

The immature seeds at 73 and 94 DAP achieved a high germination rate, estimated from the testa rupture (Figure 2). However, the protocorms formed from those immature seeds failed to differentiate and develop into plantlets (Figure 8A,(A1),B,(B1)). Eventually, the growth-arrested protocorms turned brown and withered (Figure 8(A1)). A properly developed and differentiated protocorms can form only from mature seeds (Figure 8C,(C1)), which will form a shoot at the top of the protocorm and then develop into a plantlet (Figure 8D). *Paphiopedilum* seeds require a precise mechanism to determine the timing of the germination. Based on this fact, there are two approaches for improving the current conservation strategies. The first one is to figure out what causes the growth arrest at the early protocorm stages. The second approach is to identify the major germination inhibition factors and then focus on reducing the barrier.

Many studies demonstrated that ABA has the effect of inducing and maintaining seed dormancy, and GAs can stimulate seed dormancy release and germination [14,15,31,32]. In this study, the levels of four plant hormones (ABA, GA_3_, IAA, and TZR) were determined at five different developmental stages of seed development. Our results revealed that the ABA content increased as the seed approach maturation, peaking at 14.6 ng/g. The dynamic change of ABA content in *P. armeniacum* seed was similar to that of *Rosa hybrida*, *Pinus taeda*, and other species [33,34]. There is the possibility that the high level of ABA in the early stage can inhibit the premature germination of seeds [35]. To confirm this, exogenous ABA application was performed, and the results showed that exogenous ABA treatment inhibited immature seed germination (Figure 5).

The ABA level declined after lignification was accomplished at 157 DAP, with a significant reduction in germination rate. Analysis of the relative quantification of genes showed that the content of ABA would be reduced in mature seeds of *P. armeniacum* due to the increased expression of ABA catabolized genes. This is different from the relatively high ABA content found in other mature orchid seeds, such as *Cypripedium formosanum* and *Cypripedium japonicum* [19,20]. The question is whether ABA inhibits the mature seed germination or not. The ABA biosynthesis inhibitor fluoridone was applied to the capsules once a week for a month, and the hormone analysis confirmed that the treatment effectively reduced the ABA contents from 9.4 ng/g to 2.8 ng/g. No significant differences were seen in the germination rate between the mature seeds that contained a low ABA level (2.8 ng/g) and those that contained a high ABA level (9.4 ng/g), as shown in Figure 6. This suggests that ABA does not play a significant role in repressing mature seed germination. Mature orchid seeds generally have a heavily lignified seed coat that provides protection to the embryo. Several studies attributed lignin accumulation to germination inhibition of mature seeds [21,25]. Instead of ABA, the heavily lignified seed coat formed in the mature seeds may serve as the major inhibitor for germination.

## 4. Materials and Methods

### 4.1. Plant Material and Seeds Collection

*P. armeniacum* were maintained in the greenhouse of south China botanical garden under controlled conditions with a temperature of 15~35 °C, humidity between 65%~95%, and light intensity less than 400 μmol m^−2^ s^−1^. The flowers were hand-pollinated and labeled from March to April 2019. Capsules were collected for the seed morphological measurement, asymbiotic seed germination, and hormones measurement experiments.

### 4.2. Morphological Characterization of Seeds and Embryos

For the morphological investigation, seeds were observed under stereomicroscope (Nikon SMZ745T, Tokyo, Japan) and optical microscope (Nikon E200MV, Tokyo, Japan). In order to collect information on the embryo features, the seeds were treated with 1% sodium hypochlorite for 1 h to loosen the rigid seed coat structure.

To collect detailed information about seed surface morphology characteristics, the seeds were prepared for scan electron microscopy (SEM). The seeds were mounted on stubs and coated with gold palladium in a sputter-coater JEE-420, and examined with a JSM-6360LV scanning electron microscope (JEOL, Tokyo, Japan) with a filament voltage of 15 kV. The dimensions (length and width) were measured on at least 30 seeds for every sample. Qualitative data such as seed shape, morphology, and orientation of testa cells were analyzed, and representative images were recorded.

### 4.3. Seed Asymbiotic Germination Assay

Seed capsules collected at 73, 94, 115, 136, and 157 DAP were surface sterilized by dipping into 75% (*v/v*) ethanol for 2 min, agitated for 15 min in 1 g l^−1^ mercuric chloride and 0.05% (*w/v*) Tween 20, and finally rinsed four times with sterile distilled water. The surface-disinfected capsules were cut open longitudinally, and the seeds were scooped out with sterile forceps and placed on germinating medium. Hyponex N026 medium was supplemented with 1.5 g/L activated charcoal, 2 g/L peptone, 15 g/L sucrose, and 5% coconut water were used for *P. armeniacum* germination as described preciously [21]. Germination was recognized when the embryo swelled and the testa ruptured. The germination rate is calculated from three independently grown biological replicates. Each biological replicate contained seeds collected from 10 capsules collected at the same time interval, germinated in 10 culture flasks.

### 4.4. Measurement of the Endogenous Plant Hormones Levels

Endogenous levels of ABA, GA_3_, IAA, and TZR were determined as described by [36]. Freshly harvested seeds (100 mg) at 73, 94, 115, 129, and 150 DAP were ground in liquid nitrogen, and the extraction was performed with acetonitrile solution. The QuEChERS method was used to purify the impurities. The nitrogen purge method was used to concentrate the sample.

High Performance Liquid Chromatography-Mass Spectrometry (HPLC-MS/MS, SCIEX-6500Qtrap, ABI, USA) analysis was used to detect the hormone levels. The HPLC-MS/MS was equipped with a poroshell 120 SB-C18 column (2.1 mm × 150 mm). The individual hormones and their metabolites are quantified by comparison of the measured response ratio of endogenous hormone to its internal standard and the ratio of hormone of known concentration to internal standard. The hormone content was expressed as ng g^−1^ fresh weight (FW).

### 4.5. Treatment with Exogenous ABA

To examine the effect of exogenous ABA on seed germination, 1 mL of 0.5 mg/mL ABA solution were added in the 100 mL germination medium. ABA solution was prepared with sterile water and filtered with 0.22 μm membrane.

### 4.6. Treatment with Fluridone

Ninety-four DAP seed capsules were treated with fluridone as previously described [20]. Twenty seed capsules were sprayed with 100 mL fluridone 10 μM, once a week for 1 month. After 30 days, the 124 DAP seeds were collected for germination, and ABA was analyzed as described in Section 4.3 and Section 4.4.

### 4.7. RNA Extraction, De Novo Assembly, Functional Annotation of Unigenes, and Analysis of Differential Genes Expression

One-hundred milligrams of seeds were used for RNA-seq analysis. The method for total RNA extraction was described previously [25]. The RNA purity, concentration and integrity were evaluated using agarose gel electrophoresis, Nanodrop One (Nanodrop Technologies Inc., Wilmington, DE, USA), and Agilent 2100 (Agilent Technologies Inc., Palo Alto, CA, USA). See Appendix A for the quality of the RNA used for the RNA seq analysis.

The analysis of expression level of related genes was based on the previous sequencing results of the seed transcriptome of *P. armeniacum*, under the BioProject PRJNA550294 in the National Center for Biotechnology Information (NCBI). The transcripts were assembled using Trinity v2.4.0 program with default parameters, and gene expression was estimated by applying the fragments per kilobase per million mapped reads (FPKM).

The unigenes were searched against public databases, including the non-redundant protein database (NR), Swiss-Prot, Clusters of Orthologous Groups of proteins (COG), Gene Ontology (GO), and the Kyoto Encyclopedia of Genes and Genomes Pathway (KEGG).

The libraries were constructed from seeds at five different developmental stages (66 DAP, 87 DAP, 108 DAP, 122 DAP, and 150 DAP). Pairwise comparison was performed, and DESeq2 was used to analyze the differentially expressed genes (DEGs). In the process of detecting DEGs, |log2(Fold Change)| ≥ 1 and FDR < 0.05 are used as screening criteria. Expression level from 108 DAP was taken as the middle point, and the DEGs obtained from the two comparisons (66 vs. 108, 108 vs. 150) of high overall expression were selected as a representative in the Figure 4.

### 4.8. Statistical Analysis

Statistical analysis was performed using Student’s *t*-test as indicated in the figure legends. All experiments were performed on at least three independently grown biological replicates. All values represent the mean ± SD.

## 5. Conclusions

Seed germination is a complex process controlled by multiple factors. Both chemical and physical inhibition play important roles in seed germination. The accumulation of endogenous ABA in immature *P. armeniacum* seeds may play a critical role in germination inhibition. Once the seed coat begins to lignify, the ABA level declines and the heavily lignified seed coat becomes the major germination inhibitor instead. This information present here will provide theoretical guidance for large scale propagation of *P. armeniacum*.

## Figures and Tables

**Figure 1 ijms-21-09561-f001:**
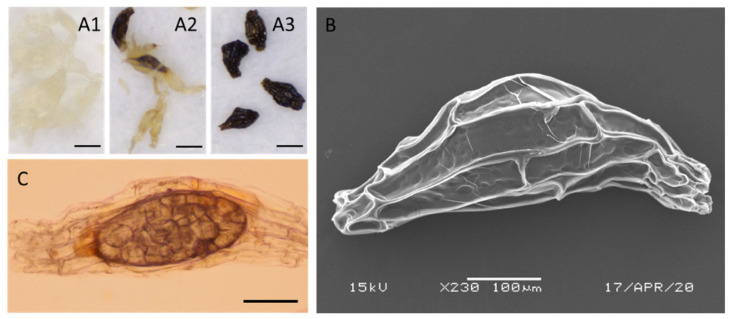
Morphological characterization of *P. armeniacum* seed and embryo. (**A1**–**A3**) Light microscope images of 73 (**A1**), 94 (**A2**), and 150 (**A3**) DAP seeds. Scale bar = 200 μm; (**B**) Scanning electron microscopy (SEM) image displays the surface feature of 150 DAP seed. Scale bar = 100 μm; (**C**) Light microscope image displays the embryo feature of 150 DAP seed, which were treated with 1% sodium hypochlorite to loosen the rigid seed coat structure. Scale Bar = 100 μm.

**Figure 2 ijms-21-09561-f002:**
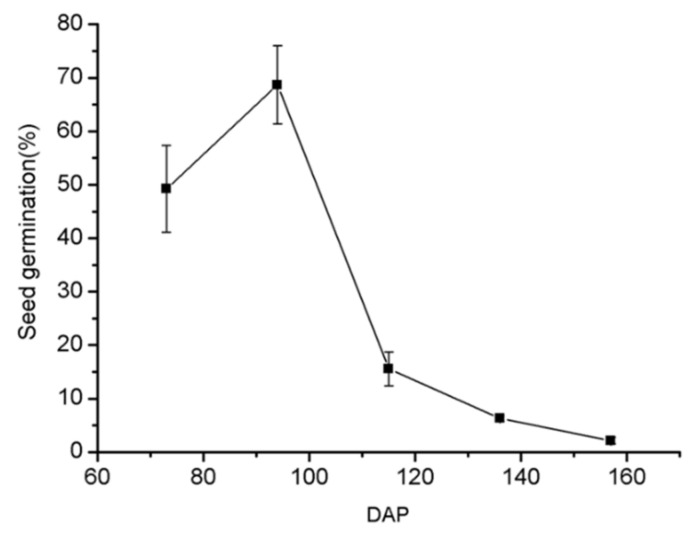
Asymbiotic seed germination rate at different developmental stages of *P. armeniacum*. *n* = 3; ±SD. Each biological replicate contained seeds collected from 10 capsules collected at the same time interval, germinated in 10 culture flasks. Seeds used for germination rate estimation were harvested at 73 DAP, 94 DAP, 115 DAP, 136 DAP, and 157 DAP.

**Figure 3 ijms-21-09561-f003:**
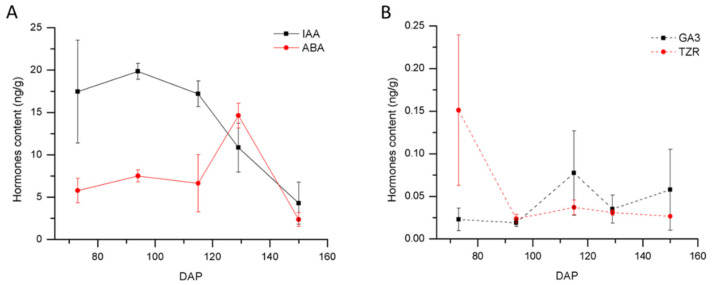
Changes of IAA (**A**), ABA (**A**), GA_3_ (**B**), and TZR (**B**) contents at different developmental stages of *P. armeniacum*. The hormone contents are expressed as ng g^−1^ fresh weight (FW). *n* = 3; ±SD. One-hundred milligrams of seeds were collected from six capsules in each biological replicate. Seeds used for hormones content analysis were harvested at 73 DAP, 94 DAP, 115 DAP, 129 DAP, and 150 DAP. The raw spectra of each hormone is now included in Appendix A.

**Figure 4 ijms-21-09561-f004:**
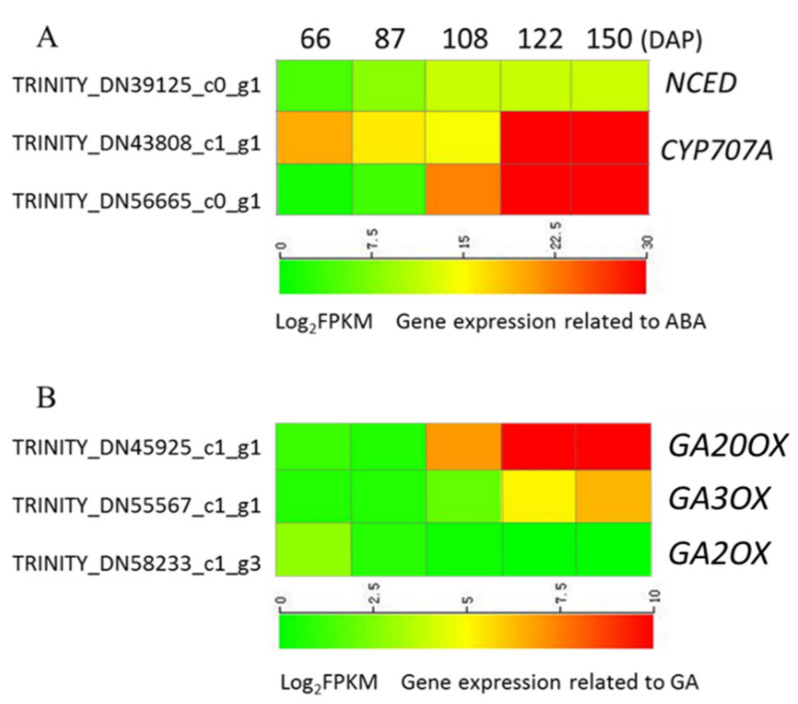
Changes in relative expression levels of ABA (**A**) and GAs (**B**) anabolic genes at different developmental stages of *P. armeniacum* seeds. TINITY_DN is the unigenes ID of related genes obtained by sequencing transcriptome; log_2_FPKM represents the expression level of the unigenes; the color represents the intensity of gene expression. The unigenes expression and sequence related to ABA and GA metabolism are listed in Appendix A.

**Figure 5 ijms-21-09561-f005:**
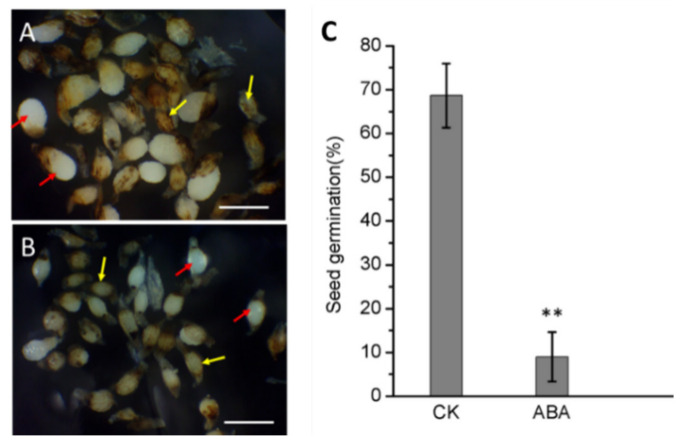
Effect of exogenous ABA on asymbiotic germination of immature seeds of *P. armeniacum*. The seeds harvested at 94 DAP were used to represent immature seeds’ germination performance. Light microscope images of (**A**) the germinated immature seeds without any treatment and the immature seeds treated with 5 μg/mL ABA (**B**) in the culture medium. The immature seeds with exogenous ABA treatment demonstrated a reduced ability on both embryo swollen and seed coat rupture, as compared to the one without ABA treatment. Red arrow: Embryo that have breakthrough seed coat; Yellow arrow: Embryo that have not broken through the seed coat. Scale bar = 1 mm. (**C**) Asymbiotic germination rate of the CK (control) and ABA-treated immature seeds. *n* = 3; ±SD; Asterisks indicate significant difference from the WT (Student’s *t* test, *p* < 0.01).

**Figure 6 ijms-21-09561-f006:**
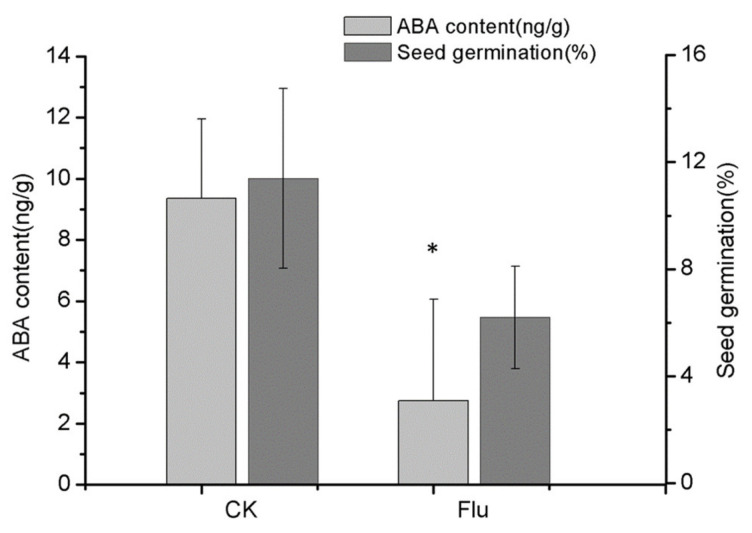
Effect of fluridone (Flu) on ABA content and asymbiotic germination of mature seed in *P. armeniacum*. The seeds were harvested at 124 DAP and used to represent the germination performance of mature seeds under reduced endogenous ABA level. *n* = 3; ±SD; an asterisk indicates a significant difference from the WT (Student’s *t* test, *p* < 0.05).

**Figure 7 ijms-21-09561-f007:**
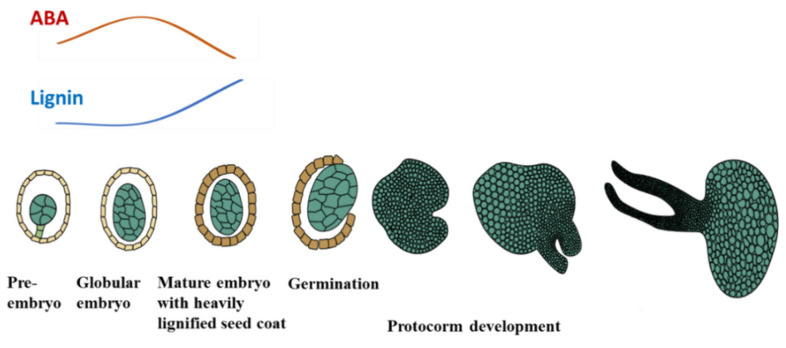
A schematic representation of embryo and protocorm development of *P. armeniacum*. *P. armeniacum* embryos developmental arrest at a stage comparable to the globular stage of *Arabidopsis* embryogenesis without obvious histodifferentiation. Seed coat lignification occurs before germination. Germination gives rise to a tubercle structure called a protocorm, from which the histodifferentiation occurs and the shoot apical meristem subsequently differentiates. The high level of ABA inhibits the germination of immature seeds. Once the lignification rises [25], the ABA level starts to decline.

**Figure 8 ijms-21-09561-f008:**
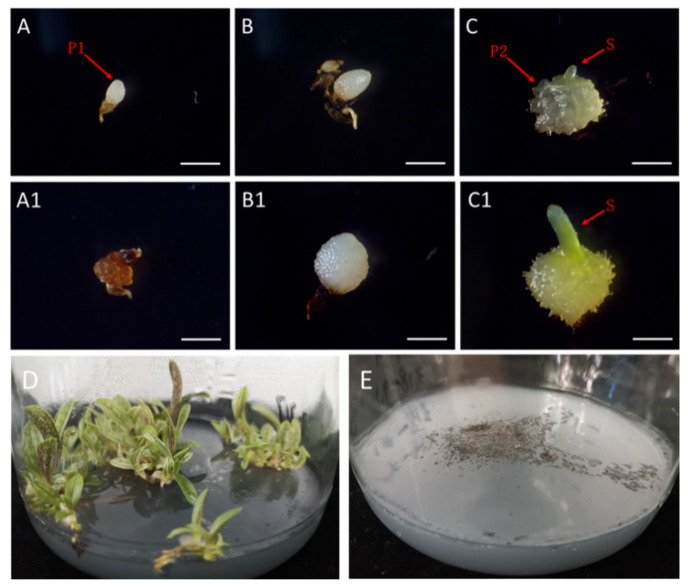
Protocorm and plantlet formation from immature and mature seeds of *P. armeniacum.* Protocorm was formed from immature seeds harvested at 73 DAP after 60 days cultivation (**A**), which failed to develop further and turned brownish after 160 days cultivation (**A1**). Protocorm was formed from immature seeds harvested at 94 DAP after 60 days cultivation (**B**), which enlarged slowly and failed to develop shoots after 140 days cultivation (**B1**). Protocorm was formed from mature seeds harvested at 124 DAP after 60 days cultivation (**C**), which rapidly enlarged and developed a shoot on top after 70 days of cultivation (**C1**). The plantlet was able to grow properly from mature seeds harvested at 124 DAP (**D**). The over-mature seeds harvested at 157 DAP failed to germinate due to the physical restriction imposed by lignin accumulation (**E**). P1: growth-arrested protocorm; P2: properly developed protocorm; S: shoot formed from the top of the protocorm; Scale bar = 0.5 mm.

**Table 1 ijms-21-09561-t001:** The size of seed and embryo at different developmental stages of *P. armeniacum*. The seed dimension was measured from SEM images, and the embryo dimension was measured from optical microscope images. *n* = 3; ±SD. The size of seed and embryo is calculated from three independently grown biological replicates. In each replicate, approximately 30 seeds were analyzed.

DAP	Seed Length (μm)	Seed Width (μm)	Seed Length/Width	Seed Color	Embryo Length (μm)	Embryo Width (μm)	Embryo Length/Width
73	524.2 ± 22.6	135.4 ± 6.1	3.9 ± 0.1	white	158.1 ± 7.9	90.0 ± 3.4	1.8 ± 0.1
94	544.2 ± 27.1	155.6 ± 6.9	3.5 ± 0.2	white/black	191.1 ± 12.4	115.3 ± 0.5	1.7 ± 0.1
115	583.0 ± 22.3	177.6 ± 10.0	3.3 ± 0.1	white/black	207.6 ± 11.8	114.4 ± 2.6	1.8 ± 0.1
129	519.5±15.1	155.7 ± 1.9	3.3 ± 0.1	black	199.5 ± 5.2	111.1 ± 0.9	1.8 ± 0.0
150	488.9 ± 6.5	172.3 ± 2.9	2.8 ± 0.0	black	216.4 ± 10.9	116.0 ± 3.3	1.9 ± 0.1

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
