# Peer review of "Abscisic Acid Inhibits Asymbiotic Germination of Immature Seeds of Paphiopedilum armeniacum"

_ijms, 2020, doi:10.3390/ijms21249561_

Round 1

Reviewer 1 Report

I have read and analyzed very carefully the manuscript entitled ‘Abscisic Acid Inhibits Asymbiotic Germination of Immature Seeds of Paphiopedilum armeniacum’. This manuscript focuses on a very interesting topic, i.e. the phenomenon of seed germination in one of the orchid species.

In my opinion this manuscript is well written but it requires improvements. Some remarks are given below.

In summary, I recommend this manuscript for publication in International Journal of Molecular Sciences after a major revision.

Remarks:

- The Authors mentioned that all the experiments were performed using three biological replicates. Please, specify how many seeds were taken for one biological replicate. Please, give that information for: Table 1, Figure 2, Figure 3, Figure 4, Figure 5, Figure 6.

- Figure 2, Figure 3: the graphs will be clearer if on the X axis seed evaluation dates are given, like in Table 1.

- The work includes the results for morphological evaluation of seeds and embryos (Table 1), seed germination (Figure 2) and changes in hormone levels (Figure 3), as well as changes in relative gene expression (Figure 4). Please, explain why for these determinations and analyses series of seeds at the same development phases were not used. 

- The description of the method used for relative gene expression needs to be completed. Please, give the weight/number of seeds taken for RNA isolation, and the isolation method used. In addition, please give the quantity of RNA taken for one experiment. Please, give the sequences of primers, genes, as in the database I cannot find the sequences for the tested genes. On what grounds do the Authors assume that unigenes, defined as TRINITY_DN39125_c0_g1 and so on, correspond to the genes associated with ABA and GA (NCED, CYP707A, GA20OX, GA3OX i GA2OX) anabolism?

- Lines 179-181; The Authors state that they have identified 3 genes undergoing differential expression, associated with ABA biosynthesis and ABA catabolism; however, in Fig. 4 they give the names of two genes (description on the right side of the figure). The results for the third gene are not described in the manuscript – please explain it.  

- Line 351; The Authors write that they were analyzing relative gene expression, but in the manuscript there is no description of the method used.

- References: please refer to ‘Instructions for Authors

Author Response

I have read and analyzed very carefully the manuscript entitled ‘Abscisic Acid Inhibits Asymbiotic Germination of Immature Seeds of Paphiopedilum armeniacum’. This manuscript focuses on a very interesting topic, i.e. the phenomenon of seed germination in one of the orchid species.

In my opinion this manuscript is well written but it requires improvements. Some remarks are given below.

In summary, I recommend this manuscript for publication in International Journal of Molecular Sciences after a major revision.

Remarks:

- The Authors mentioned that all the experiments were performed using three biological replicates. Please, specify how many seeds were taken for one biological replicate. Please, give that information for: Table 1, Figure 2, Figure 3, Figure 4, Figure 5, Figure 6.

Response: For the seed morphology characterization (Table1), approximately 30 seeds for each replicate were analyzed. Because the seeds inside the fresh capsule are dust-like, it’s difficult to get an accurate estimation of the seed number without microscopy. For the germination assay (Figure2, Figure5 and Figure 6), each biological replicate contained seeds collected from 10 capsules collected at the same time interval, germinated in 10 culture flasks. For the plant hormone (Figure 3) and gene expression analysis (Figure 4), 100 mg of seeds were collected from 6 capsules in each biological replicate. The above information was now included in the method and Figure and Table legend of the revised manuscript (Line 131, Line149, Line 185, and Line 342).

- Figure 2, Figure 3: the graphs will be clearer if on the X axis seed evaluation dates are given, like in Table 1.

Response: Seed harvest data was now included in the Figure legend of Figure 2 and Figure 3. (Line 150 and Line 186)

- The work includes the results for morphological evaluation of seeds and embryos (Table 1), seed germination (Figure 2) and changes in hormone levels (Figure 3), as well as changes in relative gene expression (Figure 4). Please, explain why for these determinations and analyses series of seeds at the same development phases were not used. 

Response: Seeds used for the relative gene expression analysisFigure 4)were harvested at 66 DAP, 87 DAP, 108 DAP, 122 DAP and 150 DAP. However, seeds used for germination (Figure 2) were harvested at 73 DAP, 94 DAP, 115 DAP, 136 DAP and 157 DAP. Seeds used for morphological evaluation of seeds and embryos (Table 1) and hormones content analysis (Figure 3) were harvested at 73 DAP, 94 DAP, 115 DAP, 129 DAP and 150 DAP. Indeed two different sets of seeds were used for different analysis due to the availability of the Paphiopedilum armeniacum.

At the beginning of the experiment, we harvest the seeds from 45 DAP to 180 DAP at 7-day interval to make sure all the key development stages were characterized and analyzed. The amounts of seeds from the exact same development phase are not enough for all the analysis, as stated in the manuscript Paphiopedilum armeniacum is a rare and endangered orchid. Considering the relatively long development stages (45 DAP- 150 DAP), we believe the seeds with a different of 7-day harvesting date have similar features in terms of gene expression and physiological features.

- The description of the method used for relative gene expression needs to be completed. Please, give the weight/number of seeds taken for RNA isolation, and the isolation method used. In addition, please give the quantity of RNA taken for one experiment. Please, give the sequences of primers, genes, as in the database I cannot find the sequences for the tested genes. On what grounds do the Authors assume that unigenes, defined as TRINITY_DN39125_c0_g1 and so on, correspond to the genes associated with ABA and GA (NCED, CYP707A, GA20OX, GA3OX i GA2OX) anabolism?

Response: The method for RNA isolation, relative gene expression and unigenes annotation is now included in the revised manuscript (Line 369 and Line376). The RNA extracted quality and quantity information is added in Figure S1. TRINITY_DN##_is the ID of every unigene from transcriptome, and the unigene definition is to align and analyze the sequence of every unigene with the database described in the method (Line 379).

- Lines 179-181; The Authors state that they have identified 3 genes undergoing differential expression, associated with ABA biosynthesis and ABA catabolism; however, in Fig. 4 they give the names of two genes (description on the right side of the figure). The results for the third gene are not described in the manuscript – please explain it.  

Response: The third unigene TRINITY_DN56665_c0_g1 also belongs to CYP707A. The detailed unigene expression is now included in Table S1.

 - Line 351; The Authors write that they were analyzing relative gene expression, but in the manuscript there is no description of the method used.

Response: The method for relative gene expression is now included in the revised manuscript (Line 383).

- Referencesplease refer to ‘Instructions for Authors

Response: Some modifications were made according to “ Instructions for Authors”.

Reviewer 2 Report

Manuscript deemed suitable after few minor corrections in the English language, and after improving the data presentation. 

Author Response

Manuscript deemed suitable after few minor corrections in the English language, and after improving the data presentation.

Response: The English writing was checked thoroughly in the revised manuscript. A significant efforts has been made to improve the data presentation.

Reviewer 3 Report

The manuscript entitled “Abscisic acid inhibit asymbiotic germination of immature seeds of Paphiopedilum armeniacum” studied the effect of ABA in asymbiotic germination of immature seeds of one of the orchid Paphiopedilum armeniacum.

Immature seeds of P. armeniacum taken out of capsules at 70-100 DAP showed relatively good germination, while fully mature seeds do not germinate. Further endogenous hormone levels were measured if they show association to their germination behaviour. ABA level was increased in a later stage, which most likely suppresses precocious germination of immature seeds. As suppression of endogenous ABA accumulation by exogenous application of fluridone did not promote germination of this orchid, it was concluded that non-germination phenotype from mature seeds would be due to physical properties from lignin accumulation rather than physiological changes by ABA.

In general, presented data look solid and interesting, however I’ve got several questions about the data and inconsistency about the description/interpretation, which need to be clarified. Also I found several things have to be corrected, which are listed below with my questions and suggestions.

  1. I agree that this study focused on the effect of ABA, but I feel the conclusion is actually “not only lignin, ABA also regulate seed germination of P. armeniacum.”  So the title could be changed?
  2. 1 In Fig1A, the legend says A2 showed seeds at 94DAP.  So at 94DAP, some seeds are already quite darkened. However, in table 1, it is mentioned that seed colour is white at 94DAP.  There should be some mistake here, I guess.
  3. Table 1. The legend says the values come from three biological replicates.   Does this mean only 3 individual seeds were used for each measurement?   If so, that is not enough. For this kind of analysis, enough numbers as population should be taken, e.g 30 – 50. Also, if I were you, having 30-50 individuals from a single capsule is a bit tricky, so I would try cut open at least 3-5 capsules at the same developmental stage (e.g. 94 DAP), and confirm the observation is trustworthy. In this case, it can be mentioned in the method section or in the legend as “around 50 seeds in total from 3 independent capsules were used the measurement.”  On the other hand, in method section (Line319), it is mentioned as “measured on at least 30 seeds”.  So the values in Table 1 are actually coming from 30 seeds instead of 3?  Also in the legend it says that “values in brackets” but there is no numbers in brackets in the table.
  4. 2. The legend says replicate number is 3.  I guess this number is for rep plates in this case, and it would be better to mention how many seeds in one replicate you had.
  5. Line 157-158. Please add citation for this statement about lignin.
  6. Line 172-173. Please add citation about a role of TZR.
  7. Line 167.   Is there any study about which bioactive GA is major in this species?  Why is only GA3 was measured here? I am not familiar with orchid, but typically GA1 or GA4 are the major ones? If hormone measurement was done by ELISA, then sometimes it is difficult to detect specific ones depending on the used antibody, but LC-MS/MS was used in this study, so I wonder why.
  8. Line 168. “GAs contain many active substances such as GA1….” This sentence is strange. It could be like “there are other bioactive gibberellins such as GA1 or GA4 besides GA3.”
  9. Gene expression section (2.4) (Line 179 -, and figure 4). There are a few GA metabolic genes mentioned here.  The gene name or enzyme name of these are GA20OX, GA3OX, and GA2OX, in italic if in case of genes (the number should not be subscript). The name refers for example “an oxidase which catalyzes oxidation of carbon 20 of GA chemical structure”, not “oxidation of GA20 (a specific compound)”.  
  10. Gene expression section. It is not clear to me if the transcriptome analysis has identified multiple NCEDs, GA20oxs,GA3oxs, and GA2oxs, but only presented ones were showing differential regulation? It is mentioned DEGs were selected based on FC is larger than 2 (above 1 in log2), but first of all, how and what all the expression level was compared against? Here, I do not expect any pairwise comparison – and I do not understand what kind of DEG they are.  
  11. Line 219. It is mentioned “ABA content was reduced in mature seeds”, but isn’t 128DAP seeds are still immature, as it was explained earlier that “as the seeds mature further, ABA level dropped” (Line 156).
  12. Figure 7. Lignin data was not shown in this study, so better to have a citation in the legend.
  13. Line 251-, Figure 8. This figure 8 was not mentioned in the results section and first appeared in discussion section. For me, it is a bit strange to show some results only in discussion so perhaps it could be mentioned in the result section as well or it could be a way to add it as supplemental figure.
  14. Line 277-278, The statement is not wrong, but I think it is clearer if it is said as “ABA does not play a significant role in repressing mature seed germination”.
  15. To me, Figure 8 is quite interesting, as I thought it would be the way to collect immature seeds around 100 DAP and let them asymbiotically germinated until I saw Figure 8. So quite important message overall would be, high percentage of germination with immature seeds can’t solve the germination problem for conservation purpose as seed germination does not lead to success in seedling establishment. So perhaps, you could also discuss something like “this study suggested the importance to focus on how to reduce the physical barrier from lignified seed coat to solve the germination problem.”

Minor points

  1. Line 165. “was formed in 94 DAP seeds” to “was formed by 94 DAP.”
  2. Line 189. Catabolized to catabolic
  3. Figure 4 legend. Line 197. “Rinity” to “Trinity”, and “ID” or “code”would be better than “number” for independent unigenes.
  4. It is minor but some sentences may need English proof-reading to improve overall quality.

Author Response

The manuscript entitled “Abscisic acid inhibit asymbiotic germination of immature seeds of Paphiopedilum armeniacum” studied the effect of ABA in asymbiotic germination of immature seeds of one of the orchid Paphiopedilum armeniacum.

Immature seeds of P. armeniacum taken out of capsules at 70-100 DAP showed relatively good germination, while fully mature seeds do not germinate. Further endogenous hormone levels were measured if they show association to their germination behaviour. ABA level was increased in a later stage, which most likely suppresses precocious germination of immature seeds. As suppression of endogenous ABA accumulation by exogenous application of fluridone did not promote germination of this orchid, it was concluded that non-germination phenotype from mature seeds would be due to physical properties from lignin accumulation rather than physiological changes by ABA.

In general, presented data look solid and interesting, however I’ve got several questions about the data and inconsistency about the description/interpretation, which need to be clarified. Also I found several things have to be corrected, which are listed below with my questions and suggestions.

 I agree that this study focused on the effect of ABA, but I feel the conclusion is actually “not only lignin, ABA also regulate seed germination of P. armeniacum.”  So the title could be changed?

Response: Both lignin and ABA contribute to the regulation of P. armeniacum seed germination. However, the majority of the data present in this manuscript is associated with the role of ABA. The detailed mechanism of lignin is ongoing project in our team. We like to publish the lignin part with more data in the future. We believe the present title is appropriate.

1.  In Fig1A, the legend says A2 showed seeds at 94DAP.  So at 94DAP, some seeds are already quite darkened. However, in table 1, it is mentioned that seed colour is white at 94DAP.  There should be some mistake here, I guess.

Response: Table 1 was corrected to white/black. And make a supplementary note, white seeds still account for the majority at 94 DAP.

Table 1. The legend says the values come from three biological replicates.   Does this mean only 3 individual seeds were used for each measurement?   If so, that is not enough. For this kind of analysis, enough numbers as population should be taken, e.g 30 – 50. Also, if I were you, having 30-50 individuals from a single capsule is a bit tricky, so I would try cut open at least 3-5 capsules at the same developmental stage (e.g. 94 DAP), and confirm the observation is trustworthy. In this case, it can be mentioned in the method section or in the legend as “around 50 seeds in total from 3 independent capsules were used the measurement.”  On the other hand, in method section (Line319), it is mentioned as “measured on at least 30 seeds”.  So the values in Table 1 are actually coming from 30 seeds instead of 3?  Also in the legend it says that “values in brackets” but there is no numbers in brackets in the table.

Response: The values in Table 1 are coming from 30 seeds ×3. The Table legend was adjusted for clarification (Line 130).

3. The legend says replicate number is 3.  I guess this number is for rep plates in this case, and it would be better to mention how many seeds in one replicate you had.

Response: The number of seeds is not included in the revised manuscript (Line 149).

4.Line 157-158. Please add citation for this statement about lignin.

Response: The citation was included in the revised manuscript (line 166).
Line 172-173. Please add citation about a role of TZR .

Response: The citation was included in the revised manuscript (line 181).
Line 167.   Is there any study about which bioactive GA is major in this species?  Why is only GA3 was measured here? I am not familiar with orchid, but typically GA1 or GA4 are the major ones? If hormone measurement was done by ELISA, then sometimes it is difficult to detect specific ones depending on the used antibody, but LC-MS/MS was used in this study, so I wonder why.

Response: Considering the unique orchid seed structure, the previous results from model plants might not apply to P. armeniacum seed. GA3 has been reported to impact orchid seed germination of Phalaenopsis orchids. According to this, we initially target GA3 for the hormone profiling.

Reference: KIA, H.H., Onsinejad, R. and Yari, F., 2015. The effect of pollination time and gibberellic acid (GA3) on the production and seed germination of Phalaenopsis orchids.

The hormone measurement was done by LC-MS/MS in this study. We also tried ELISA measurement at first, but the sensitivity is not high enough to detect GA contents.
Line 168. “GAs contain many active substances such as GA1….” This sentence is strange. It could be like “there are other bioactive gibberellins such as GA1 or GA4 besides GA3.”

Response: The statement is adjusted according to the reviewer’s suggestion (Line 176).
Gene expression section (2.4) (Line 179 -, and figure 4). There are a few GA metabolic genes mentioned here.  The gene name or enzyme name of these are GA20OX, GA3OX, and GA2OX, in italic if in case of genes (the number should not be subscript). The name refers for example “an oxidase which catalyzes oxidation of carbon 20 of GA chemical structure”, not “oxidation of GA20 (a specific compound)”.  

Response: The modification was made in the revised manuscript (Line 201 and Line 207).
Gene expression section. It is not clear to me if the transcriptome analysis has identified multiple NCEDs, GA20oxs, GA3oxs, and GA2oxs, but only presented ones were showing differential regulation? It is mentioned DEGs were selected based on FC is larger than 2 (above 1 in log2), but first of all, how and what all the expression level was compared against? Here, I do not expect any pairwise comparison – and I do not understand what kind of DEG they are.  

Response: Transcriptome analysis has identified multiple NCEDs, GA20oxs, GA3oxs, and GA2oxs ( the full list is now included in Table S1).  The method for unigenes annotation and analysis of differential gene expression is now included in the revised manuscript (Line 383). Differentially expressed genes (DEGs) between libraries were identified as those with the fold change (FC) of the expression level (FC≥ 1 or FC≤0.5 under P-value ≤0.05), FDR ≤0.05). Only the four adjacent periods that meet the DEG standard were selected.

Line 219. It is mentioned “ABA content was reduced in mature seeds”, but isn’t 128DAP seeds are still immature, as it was explained earlier that “as the seeds mature further, ABA level dropped” (Line 156).

Response:  As stated in line 102, the seed reached maturity at 115 DAP. So 128 DAP seeds are fully matured.
Figure 7. Lignin data was not shown in this study, so better to have a citation in the legend.

Response: The reference is now included in Figure 7 legend.
Line 251-, Figure 8. This figure 8 was not mentioned in the results section and first appeared in discussion section. For me, it is a bit strange to show some results only in discussion so perhaps it could be mentioned in the result section as well or it could be a way to add it as supplemental figure.

Response: Figure 8 describe the protocorm and plantlet formation, which indeed was not mentioned in the results part. However, we believe the information serves as an extension for the difference developmental pattern of immature and mature seeds. We believe it’s appropriate to stay in the discussion part.
Line 277-278, The statement is not wrong, but I think it is clearer if it is said as “ABA does not play a significant role in repressing mature seed germination”.

Response: The statement was corrected according to the reviewer’s suggestion.
To me, Figure 8 is quite interesting, as I thought it would be the way to collect immature seeds around 100 DAP and let them asymbiotically germinated until I saw Figure 8. So quite important message overall would be, high percentage of germination with immature seeds can’t solve the germination problem for conservation purpose as seed germination does not lead to success in seedling establishment. So perhaps, you could also discuss something like “this study suggested the importance to focus on how to reduce the physical barrier from lignified seed coat to solve the germination problem.”

Response: We believe there are two approaches for improving the current conservation strategies. The first one is to figure out what cause the growth arrest at the early protocorm stages. The second approach is to focus on reducing the physical barrier from lignified seed coat as suggested by the reviewer.

Minor points
Line 165. “was formed in 94 DAP seeds” to “was formed by 94 DAP.”
Line 189. Catabolized to catabolic
Figure 4 legend. Line 197. “Rinity” to “Trinity”, and “ID” or “code”would be better than “number” for independent unigenes.
It is minor but some sentences may need English proof-reading to improve overall quality.

Response: The above points were corrected according to the reviewer’s suggestion. The English writing was checked thoroughly in the revised manuscript.

Reviewer 4 Report

This paper present an interesting study regarding the influence of endogenous hormone – abscisic acid on the germination of P. armeniacum seeds, which brings new knowledge in this field.

The manuscript is well written, but some revisions are required:

In section Results, paragraph 2.3 supplementary results must be presented regarding the HPLC-MS/MS determination of endogenous hormones during seed germination, not only the determined content of those hormones. Mass spectra and chromatograms would be appropriate.

In section Materials and methods, paragraph 4.4 more details must be given regarding the experimental protocol used for quantitative determination of HPLS-MS/MS of endogenous plant hormones.

In section Materials and methods, paragraph 4.7, more information regarding the analysis of expression level of related genes would be useful for better understating of the applied method.

 Conclusions could be more detailed, in order to reflect all the applied studies, as well as the applicability of this study in establishing effective germination protocols for P. armeniacum seeds

Author Response

This paper present an interesting study regarding the influence of endogenous hormone – abscisic acid on the germination of P. armeniacum seeds, which brings new knowledge in this field.

The manuscript is well written, but some revisions are required:

In section Results, paragraph 2.3 supplementary results must be presented regarding the HPLC-MS/MS determination of endogenous hormones during seed germination, not only the determined content of those hormones. Mass spectra and chromatograms would be appropriate.

Response: The spectra of endogenous hormones is now included in Figure S2.

In section Materials and methods, paragraph 4.4 more details must be given regarding the experimental protocol used for quantitative determination of HPLS-MS/MS of endogenous plant hormones.

Response: The detailed methods on HPLC-MS/MS is now included in the method (Line 352).

In section Materials and methods, paragraph 4.7, more information regarding the analysis of expression level of related genes would be useful for better understating of the applied method.

Response: The detailed methods on gene expression analysis is now included in the method (Line 383).

Conclusions could be more detailed, in order to reflect all the applied studies, as well as the applicability of this study in establishing effective germination protocols for P. armeniacum seeds

Response: Additional statement is now added in the conclusion (line 396).

Round 2

Reviewer 3 Report

In this revised manuscript, the authors tried to answer and addressed the questions raised by reviewers. 

I’m mostly happy with this version, but I still have a few things to clarify.

In the previous version, I’ve asked why GA3 was measured not GA1 or GA4, and the authors’ answer was “GA3 has been reported to impact orchid seed germination” with a reference.  To my knowledge, this paper does not mention any endogenous hormone levels, and they simply applied GA3 exogenously.  In this case, regardless which bioactive GA was given, in most cases one would see promotion of germination, simply because GA3 is also bioactive GA (although effectiveness could vary – you might need more concentration to see the same level of promotion). We know GA4 is major in Arabidopsis and GA1 in rice, but if you give GA3, both species also respond. Therefore, to me the initial target based on the study/paper does not really make any sense, unfortunately.  However, this paper is not focused on GA, and I think it is ok for now.

I still do not understand the explanation of DEGs in transcriptome. The method section was added, but it says DEGs between libraries – so each pairwise comparison between the various timepoints was performed? For example, using DESeq2 or EdgeR or any other R programs?

In the time course analysis, for example you set the earliest time point as control (in this study 66DAP), and compare all the other time points against 66DAP, then you can easily follow how the gene expression pattern changes and calculate a fold-change against “the control”.  But essentially, given the data of log2 FPKM (or even simple FPKM), you have expression pattern of limited numbers of potential hormone metabolic unigenes, without any DEG selection, perhaps you simply can mention, some genes showed higher/lower expression at specific time points based on the FPKM/log2 FPKM values? Based on the supplementary table with expression values (look like they are log2 FPKM), some of the genes not selected in the Figure 4 showed different pattern with FC bigger than 1 or smaller than -1. So I’m a bit lost what was explained.  I guess it is reasonably ok to mention something like “this unigene for NCED showed increased expression at the late maturation period, therefore it may play a role in increased accumulation of ABA in seed maturation( Fig 3).”  This does not require any DEG selection, and much more straight forward based on identified expression pattern.

My comment/suggestion about discussion how to truly solve the germination problem (related to Fig 8), the authors have given me a great answer, and it is a pity that that was not included in the discussion.  I really think the conclusive statement could have given a clear direction of the future research.

Author Response

Response to Reviewer3 (round3)

In this revised manuscript, the authors tried to answer and addressed the questions raised by reviewers. 

I’m mostly happy with this version, but I still have a few things to clarify.

In the previous version, I’ve asked why GA3 was measured not GA1 or GA4, and the authors’ answer was “GA3 has been reported to impact orchid seed germination” with a reference.  To my knowledge, this paper does not mention any endogenous hormone levels, and they simply applied GA3 exogenously.  In this case, regardless which bioactive GA was given, in most cases one would see promotion of germination, simply because GA3 is also bioactive GA (although effectiveness could vary – you might need more concentration to see the same level of promotion). We know GA4 is major in Arabidopsis and GA1 in rice, but if you give GA3, both species also respond. Therefore, to me the initial target based on the study/paper does not really make any sense, unfortunately.  However, this paper is not focused on GA, and I think it is ok for now.

I still do not understand the explanation of DEGs in transcriptome. The method section was added, but it says DEGs between libraries – so each pairwise comparison between the various timepoints was performed? For example, using DESeq2 or EdgeR or any other R programs?

Response: Yes, pairwise comparison between library constructed from samples collected between the various timepoints was. DESeq2 was used to analyze the different expression genes (DEGs). In the process of detecting differentially expressed genes, |log2(Fold Change)|≥1 and FDR<0.05 were used as screening criteria. We are sorry for the confusion, and the detail is now included in the revised manuscript (Line 389-392).

In the time course analysis, for example you set the earliest time point as control (in this study 66DAP), and compare all the other time points against 66DAP, then you can easily follow how the gene expression pattern changes and calculate a fold-change against “the control”.  But essentially, given the data of log2 FPKM (or even simple FPKM), you have expression pattern of limited numbers of potential hormone metabolic unigenes, without any DEG selection, perhaps you simply can mention, some genes showed higher/lower expression at specific time points based on the FPKM/log2 FPKM values? Based on the supplementary table with expression values (look like they are log2 FPKM), some of the genes not selected in the Figure 4 showed different pattern with FC bigger than 1 or smaller than -1. So I’m a bit lost what was explained.

Response: That shown in the supplementary table was the expression level of related genes (FPKM). We not only pay attention to the expression of ABA and GAs related genes to reflect the changes in ABA and GA level, but also want to find out which DEGs contribute to the different development stages of seeds in this process. Therefore, 108DAP was taken as the middle point, and the DEGs obtained from the two comparisons (66vs108, 108vs150) (related to ABA: TRINITY_DN39125_c0_g1TRINITY_DN43808_c1_g1, TRINITY_DN56665_c0_g1, TRINITY_DN47606_c1_g1 related to GAs: TRINITY_DN45925_c1_g1TRINITY_DN59645_c0_g1TRINITY_DN55567_c1_g1TRINITY_DN58233_c1_g3). The high overall expression of each gene was selected as a representative in the Figure 4. (line391-394, supplementary table1)

 I guess it is reasonably ok to mention something like “this unigene for NCED showed increased expression at the late maturation period, therefore it may play a role in increased accumulation of ABA in seed maturation(Fig 3).”  This does not require any DEG selection, and much more straight forward based on identified expression pattern.

Response: Directly using the expression of related genes (FPKM) can indeed show the overall trend more intuitively, as shown below figure. But we also want to know which unigene related to ABA and GAs was important for the seed development, so the DEGs were selected shown in Figure4.

My comment/suggestion about discussion how to truly solve the germination problem (related to Fig 8), the authors have given me a great answer, and it is a pity that that was not included in the discussion.  I really think the conclusive statement could have given a clear direction of the future research.

Response: This statement is now included in the revised manuscript (line 269-271).

Reviewer 4 Report

I consider that the authors have responded appropriately to the problems previously reported by me, the manuscript being improved. Accordingly, I recommend this material for publication.

Author Response

Thank the of affirmation of the reviewer.